

# Exploring the antioxidant potential of chalcogen-indolizines throughout *in vitro* assays

Cleisson Schossler Garcia[1], Marcia Juciele da Rocha[1], Marcelo Heinemann Presa[1], Camila Simões Pires[1], Evelyn Mianes Besckow[1], Filipe Penteado[2], Caroline Signorini Gomes[3], Eder João Lenardão[3], Cristiani Folharini Bortolatto[1] and César Augusto Brüning[1]

[1] Laboratory of Biochemistry and Molecular Neuropharmacology (LABIONEM), Chemical, Pharmaceutical and Food Sciences Center (CCQFA), Federal University of Pelotas, Pelotas, Rio Grande do Sul, Brazil
[2] Department of Chemistry, Federal University of Santa Maria, Santa Maria, Rio Grande do Sul, Brazil
[3] Laboratory of Clean Organic Synthesis (LASOL), Chemical, Pharmaceutical and Food Sciences Center (CCQFA), Federal University of Pelotas, Pelotas, Rio Grande do Sul, Brazil

Corresponding authors
Cristiani Folharini Bortolatto,
cbortolatto@gmail.com
César Augusto Brüning,
cabruning@yahoo.com.br

## ABSTRACT

Reactive oxygen species (ROS) and reactive nitrogen species (RNS) are highly reactive molecules produced naturally by the body and by external factors. When these species are generated in excessive amounts, they can lead to oxidative stress, which in turn can cause cellular and tissue damage. This damage is known to contribute to the aging process and is associated with age-related conditions, including cardiovascular and neurodegenerative diseases. In recent years, there has been an increased interest in the development of compounds with antioxidant potential to assist in the treatment of disorders related to oxidative stress. In this way, compounds containing sulfur (S) and/or selenium (Se) have been considered promising due to the relevant role of these elements in the biosynthesis of antioxidant enzymes and essential proteins with physiological functions. In this context, studies involving heterocyclic nuclei have significantly increased, notably highlighting the indolizine nucleus, given that compounds containing this nucleus have been demonstrating considerable pharmacological properties. Thus, the objective of this research was to evaluate the *in vitro* antioxidant activity of eight S- and Se-derivatives containing indolizine nucleus and different substituents. The *in vitro* assays 1,1-diphenyl-2-picryl-hydrazil (DPPH) scavenger activity, ferric ion ($Fe^{3+}$) reducing antioxidant power (FRAP), thiobarbituric acid reactive species (TBARS), and protein carbonylation (PC) were used to access the antioxidant profile of the compounds. Our findings demonstrated that all the compounds showed FRAP activity and reduced the levels of TBARS and PC in mouse brains homogenates. Some compounds were also capable of acting as DPPH scavengers. In conclusion, the present study demonstrated that eight novel organochalcogen compounds exhibit antioxidant activity.

## INTRODUCTION

Some illnesses and neurological processes, such as cancer (*Huber et al., 1991*), aging (*Cadenas & Davies, 2000*), and inflammation (*Valério et al., 2009*) share a common link with oxidative stress playing a pivotal role in their pathophysiology, suggesting that oxidative stress can be a key factor or significant contributor to the exacerbation of these diseases.

When formed in high concentrations, reactive oxygen species (ROS), which are electrophilic molecules produced physiologically, can react with nucleic acids, lipids, and proteins, resulting in oxidative damage to these macromolecules (*Juan et al., 2021*). Both ROS and reactive nitrogen species (RNS) can cause oxidative and nitrative stress, leading to cellular damage.

The brain is vulnerable to excessive oxidative insults because of its abundant lipid content, high energy requirements, and weak antioxidant capacity (*Lee, Cha & Lee, 2020*). These fatty acids act as a source for decomposition reactions such as lipid peroxidation, wherein a single free radical can trigger the breakdown of neighboring molecules. Of note, polyunsaturated fatty acids are significant biological targets susceptible to oxidative damage caused by ROS.

Cells have evolved various defense mechanisms to mitigate the damaging effects of both ROS and RNS, including antioxidant enzymes and molecules that scavenge ROS and RNS and repair oxidative/nitrative damage. These cellular defense mechanisms help to maintain the cell's natural redox state and prevent the accumulation of harmful ROS and RNS (*Schieber & Chandel, 2014*). However, when a transient or long-term increase in steady-state ROS levels occurs, it can disrupt cellular metabolic and signaling pathways, particularly those dependent on ROS, leading to oxidative modifications of the organism's macromolecules. If not counterbalanced, these modifications can ultimately lead to cell death through necrosis or apoptosis. From these points, the oxidative stress process is established (*Lushchak & Storey, 2021*).

In recent years, there has been a surge in interest in using antioxidants to treat disorders caused by oxidative stress (*Nogueira & Rocha, 2010*). In this context, special consideration should be paid to chalcogen-indolizines and their derivatives. Sulfur (S) is required for the formation of several enzymes and antioxidant agents, such as glutathione and thioredoxin. Some sulfur-containing compounds can effectively neutralize ROS and RNS, and sulfur is also well-known for its role in managing pathological diseases caused by oxidative stress (*Manna, Das & Parames, 2013*; *Rezk et al., 2004*).

Similarly, selenium (Se) is a trace element embedded in selenoproteins and essential to physiological and biochemical functions (*Lenardao, Santi & Sancineto, 2018*). Furthermore, selenium-containing compounds have been reported as potent antioxidants, possibly due to their ability to mimic enzymes such as dehydroascorbate reductase (DHAR), glutathione-S-transferase (GST), and glutathione peroxidase (GPx), as well as by serving as a substrate for thioredoxin reductase (TrxR) (*Nogueira, Zeni & Rocha, 2004*; *Palmieri & Sblendorio, 2007*; *Prigol et al., 2009*; *Anghinoni et al., 2023*).

Within the whole class of *N*-heterocycles, the indolizine core has a set of properties that allow a diverse range of applications in both medicinal and materials chemistry (*Sadowski, Klajn & Gryko, 2016*). Despite their low occurrence in nature, synthetic derivatives have demonstrated significant biological activity as phosphatase inhibition (*Bender & Beavo, 2006*), antimicrobial (*Dawood & Abbas, 2020*), anticonvulsant and anti-inflammatory activities (*Dawood et al., 2006*; *Shrivastava et al., 2017*), anti-schizophrenia (*Xue et al., 2016*), calcium entry blockers (*Poty et al., 1994*), antioxidant properties (*Moradi & Kohi, 2020*; *Uppar et al., 2020*), among others.

Therefore, it is interesting to explore hybrid compounds for the development of new drugs that can provide support in treating illnesses caused by oxidative stress. Considering that even minor changes in molecular structures can significantly impact a drug's effectiveness, novel sulfur- and selenium-containing indolizine hybrid compounds with different chemical substituents have been synthesized. Many of the characteristics of sulfur- and selenium-containing compounds are associated with their antioxidant activity. Our research group has previously demonstrated the synthesis and ABTS$^{\bullet+}$ scavenger activity of certain chalcogen-indolizines (*Penteado et al., 2019*). Inspired by the preliminary outcomes, this study aimed to investigate the antioxidant potential of eight sulfur- or selenium-containing compounds through several *in vitro* assays.

## MATERIALS & METHODS

### Animals

Some *in vitro* analyses were carried out using the brain of male Swiss mice (25–30 g). The animals were obtained from the Central Bioterium of Federal University of Pelotas (UFPel). They were kept in a room at $22 \pm 1\,°C$, with photoperiod 12 h light/dark cycle with lights on at 7:00 a.m. and free access to access to regular laboratory chow pellets and water and they were housed in $20 \times 30 \times 13$ cm cages (five animals/cage). The animal experiments were performed in according with the NIH guidelines for the care and use of laboratory animals (NIH publications no 8023, revised 1978) and all manipulations were conducted according to the rules of the UFPel Animal Ethics and Welfare Committee (CEEA 12231-2019). All efforts were made to minimize animal suffering and to reduce the number of animals used. For animal euthanasia, each mouse was exposed to cotton soaked with isoflurane (anesthetic agent) in a closed box until cardiorespiratory arrest was observed. After that, the brain was dissected. The study used a total of 48 mouse brains to perform lipid peroxidation and protein carbonylation assays. Specifically, 24 were designated for the first assay, and the remaining samples were allocated for the second assay. The experiments were carried out three times in duplicate, in line with the literature's recommendation to conduct the experiment a minimum of three times on separate occasions (*Carraro Junior et al., 2021*; *Hartwig de Oliveira et al., 2020*). Rodents are often used as animal models to investigate various diseases, such as neurodegenerative diseases and mood disorders. Furthermore, they are used to clarify the underlying pathophysiology, such as the oxidative stress commonly associated with these conditions (*Balmus et al., 2016*; *Zhang et al., 2017*). In this sense, the use of animal tissue in investigating the antioxidant properties of new

compounds is well accepted in the literature (*Chagas et al., 2015*; *Gomes et al., 2020*; *Vogt et al., 2018*).

## Chemicals

The thioindolizines 2-phenyl-1-(phenylthio)indolizine (SIN-1), 2-(4-chlorophenyl)-1-(phenylthio)indolizine (SIN-2), 1-(phenylthio)-2-(*p*-tolyl)indolizine (SIN-3-), 2-(4-chlorophenyl)-1-((4-fluorophenyl)thio)indolizine (SIN-4), and 2-(4-chlorophenyl)-1-((4-methoxyphenyl)thio)indolizine (SIN-5), and the selenoindolizines 2-phenyl-1-(phenylselanyl)indolizine (SeI), 2-(4-chlorophenyl)-1-(phenylselanyl)indolizine (ClSeI), and 1-(phenylselanyl)-2-(*p*-tolyl)indolizine (MeSeI) (Fig. 1), were synthesized in the Laboratory of Clean Organic Synthesis (LASOL), located at UFPel, according to *Penteado et al. (2019)* and the purity (>99%) and the structure of the compounds were determined by GC-MS and by $^1$H and $^{13}$C-NMR (nuclear magnetic resonance) analysis (Supplementary Material), respectively. The obtained data were in perfect agreement with those from the literature (*Penteado et al., 2019*). 1,1-Diphenyl-2-picryl-hydrazil (DPPH), sodium nitroprusside (SNP), thiobarbituric acid (TBA), potassium phosphate monobasic ($KH_2PO_5$), potassium persulfate ($K_2S_2O_8$), 2-amino-2-hydroximethyl-propane-1,3-diol hydrochloride (Tris-HCl), potassium phosphate buffer (TFK), sodium phosphate buffer, dinitrophenyl hydrazine (DNPH), triazine, iron(III) chloride ($FeCl_3$), ascorbic acid (AA), and trolox (Trlx) were purchased from Sigma Chemical Co. (St Louis, MO, USA). All additional reagents were of analytical grade and obtained from standard commercial suppliers. The compounds were dissolved in dimethyl sulfoxide (DMSO).

## Free radical scavenging activity (DPPH) assay

The compounds were tested for their ability to scavenge free radicals by the 2,2-diphenyl-1-picrylhydrazyl (DPPH) assay following the previously described method (*Sharma & Bhat, 2009*). The compounds to be evaluated were added to different tubes at concentrations ranging from 10 to 500 µM (10 µl) and AA (10 µl), as a positive control, at the same concentrations of the compounds. The synthetic radical solution was prepared using methanol, resulting in a dark purple color, and was then pipetted into each tube (1 mL, 50 µM), and the solution was incubated in the dark for 30 min at 25 °C in water bath (Quimis, Q304M-2105). A spectrophotometer (Bel, V-M5) and cuvettes were used to measure the decrease in absorbance, at 517 nm. Each experiment was carried out three times in duplicate. The results were expressed as a percentage of the vehicle.

## Ferric reducing antioxidant power ferric reducing antioxidant power (FRAP) assay

This assay assesses the capacity of the compound to reduce the ion Fe-III to Fe-II in an aqueous solution containing the 2,4,6 tripyridyltriazine (TPTZ) ligand (*Yoshino & Murakami, 1998*). The absorbance measurements of the generated colored ferrous complex (Fe-II/TPTZ) correspond to the reducing capability of the studied compounds. The ferric reducing antioxidant power (FRAP) reagent was obtained by combining 10:1:1 proportions of 38 mM anhydrous sodium acetate in distilled water (pH 3.8), 20 mM $FeCl_3.6H_2O$ in distilled water, and 10 mM TPTZ in 40 mM HCl. Before each experiment, this reagent was

**Figure 1 Chemical structures of thio and selenoindolizines.** SIN-1: 2-phenyl-1-(phenylthio)indolizine; SIN-2: 2-(4-chlorophenyl)-1-(phenylthio)indolizine; SIN-3: 1-(phenylthio)-2-(p-tolyl)indolizine; SIN-4: 2-(4-chlorophenyl)-1-((4-fuorophenyl)thio)indolizine; SIN-5: 2-(4-chlorophenyl)-1-((4-methoxyphenyl)thio)indolizine; SeI: 2-phenyl-1-(phenylselanyl)indolizine; ClSeI: 2-(4-chlorophenyl)-1-(phenylselanyl)indolizine; MeSeI: 1-(phenylselanyl)-2-(p-tolyl)indolizine.

freshly prepared. The compounds (1–25 µM) and AA (1–25 µM) (10 µL) were incubated with the FRAP solution (990 µL) for 40 min at 37 °C in a water bath (Quimis, Q304M-2105) in the dark. The absorbance was measured at 593 nm in a spectrophotometer (Bel, V-M5). Each experiment was carried out three times in duplicate. The results were expressed in absorbance.

## Lipid peroxidation assay

The compounds were dissolved in DMSO and used at concentrations of 1 to 10 µM in different tubes for the TBARS analysis, following the previously described protocol (*Ohkawa, Ohishi & Yagi 1979*). To induce lipid peroxidation in the mouse brain homogenate, SNP (50 mM) was used in all groups, except in the control one. With the aid of a Potter homogenizer, the tissue was homogenized in 50 mM Tris-HCl buffer, pH 7.4, at a ratio of 1:10 (w/v). Then, it was centrifuged at $2,500\times$ g (Kasvi, K14-0815CX), and the supernatant (100 µL) was mixed with a solution prepared with 50 mM Tris-HCl buffer pH 7.4 (30 µL), and SNP 0.3 mM (50 µL). Additionally, 10 µL of DMSO were added to the vehicle group, and for the tested concentrations, 10 µL of the diluted compound for each concentration were added. Water Milli-Q was added to complete 300 µL o final volume. The sample was then incubated at 37 °C for 1 h in a water bath (Quimis, Q304M-2105). Afterward, 0.8% TBA (500 µL), acetic acid buffer pH 3.4 (500 µL), and sodium dodecyl sulfate 8.1% (SDS) (200 µL) were added. Finally, after incubating at 95 °C for 1 h in a water bath (Solab, SL - 150/10), the absorbance was measured using cuvette and

spectrophotometer (Bel, V-M5) at 532 nm. Furthermore, Trlx was employed as a positive control at a range concentration of 1 to 100 µM. Each experiment was repeated three to five times and performed in duplicate. Results were expressed as nmol TBARS/g of tissue.

## Protein carbonylation assay

The carbonyl content of proteins in the mouse brain was assessed using a modified method (*Levine et al., 1990*). This approach considers the damage caused by oxidative stress. Briefly, with a Potter homogenizer, brain homogenates were prepared with Tris-HCl buffer (50 mM; pH 7.4) in a proportion 1:10 (w/v) and subsequently the samples were diluted 1:8 (Tris-HCl buffer, 50 mM, pH 7.4). The compounds at different concentrations (10–100 µM) were tested. In different test tubes, aliquots of 940 µL of homogenate dilutions were incubated at 37 °C in a water bath (Quimis, Q304M-2105) for 2 h in the presence of 10 µL of the compounds (10–100 µM) and 50 µL of SNP (1 mM) to stimulate protein carbonyl production. After the incubation, 200 µL of DNPH (10 mM in 2.0 M HCl) or 200 µL of 2.0 M HCl (in the blank tubes) was added in the tubes. The tubes were incubated for 1 h in the dark at room temperature and stirred every 15 min, to facilitate the reaction. Next, 500 µL of denaturation buffer (sodium phosphate buffer, 150 mM, pH 6.8 containing 3% SDS), 1.5 mL of ethanol, and 1.5 mL of hexane were added to all tubes. They were shaken for 40 s with a vortex mixer and centrifuged (Kasvi, K14-0815CX) for 15 min at $2,500 \times$ g. After the centrifugation, the supernatant was carefully removed, and the pellet obtained was separated washing two times with one mL of ethanol/ethyl acetate solution (1:1, v/v). After 2 min drying period at room temperature, the pellet was dissolved in one mL of denaturing buffer under stirring, and the absorbance was measured on a spectrophotometer (Bel, V-M5) at 370 nm. Trolox (10–100 µM) was used as a positive control in the same conditions. Each experiment was repeated three times and performed in duplicate for each concentration of the compounds and in uniplicate for blank tubes without DNPH. The final absorbance was calculated by the mean of the duplicate of each concentration minus the absorbance of the blank. The results were expressed as nmol carbonyl/g of tissue.

## Statistical analysis

The statistical analyses were performed in GraphPadPrism v.8.0.2 software. The normality of the data was evaluated by D'Agostino-Pearson test. One-way analysis of variance (ANOVA) followed by the Newman–Keuls *post hoc* test was employed. All experiments are presented as the mean ± standard error of the mean (SEM). Values of $p < 0.05$ were considered statistically significant.

## RESULTS

### Radical scavenging activity (DPPH) assay

SeI and SIN-1 showed DPPH radical scavenging activity at concentrations equal or greater than 100 µM ($F_{(5,12)} = 34.98$; $p < 0.0001$; $F_{(5,12)} = 17.61$; $p < 0.0001$, respectively). On the other hand, the data showed that ClSeI, MeSeI, SIN-3 and SIN-4 had a scavenging activity in concentrations equal to or higher than 300 µM ($F_{(5,12)} = 7.98$; $p < 0.01$; $F_{(5,12)} = 13.31$;

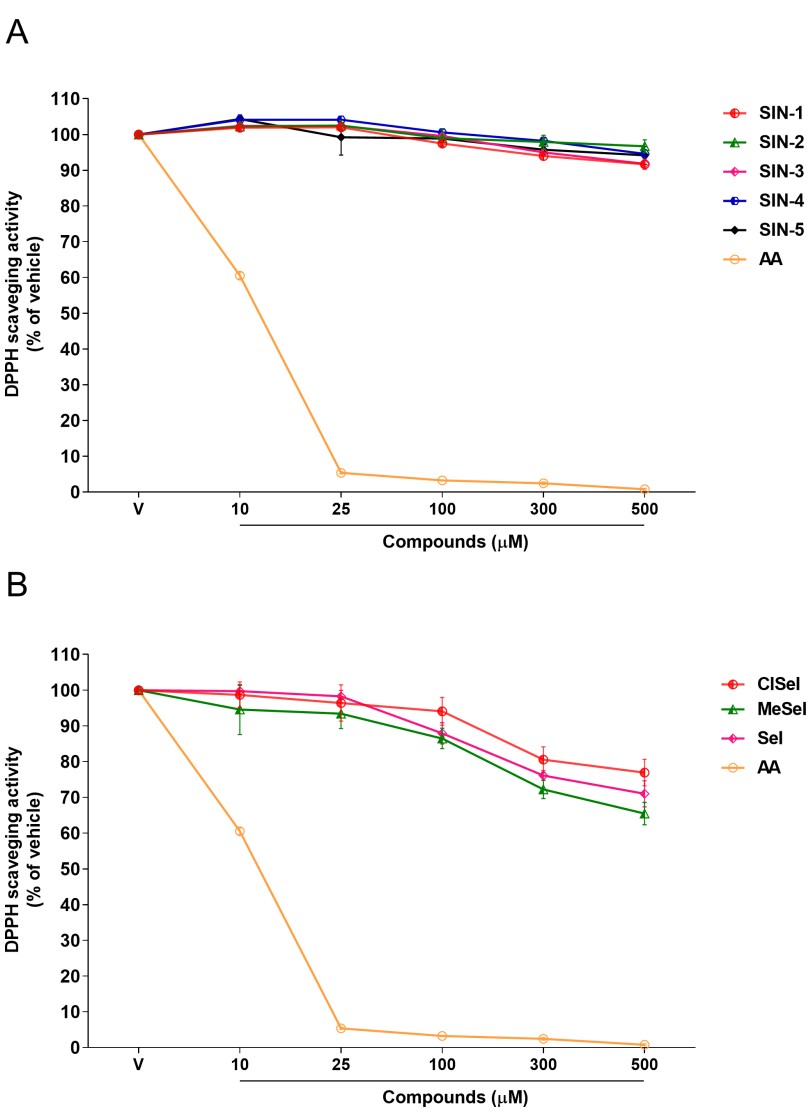

**Figure 2** **Effects of (A) thioindolizines and (B) selenoindolizines on DPPH radical capture test.** As a positive control, ascorbic acid (AA) was employed. The data are expressed in mean ± SEM of three distinct experiments conducted in duplicate. The results are presented as a percentage of the vehicle (V) group. DPPH: 1,1-diphenyl-2-picryl-hydrazil.

$p < 0.001$; $F_{(5,12)} = 32.19$; $p < 0.0001$; $F_{(5,12)} = 13.41$; $p < 0.001$, respectively). The compounds SIN-2 and SIN-5 did not show significant scavenging activity of DPPH radical ($F_{(5,12)} = 2.08$; $p = 0.138$; $F_{(5,12)} = 2.35$; $p = 0.104$, respectively). The positive control AA was effective in the radical-scavenging activity in all concentrations tested when compared with the vehicle group ($F_{(5,12)} = 8036$; $p < 0.0001$) (Fig. 2).

## Ferric reducing activity

One-way ANOVA analysis revealed that SeI, ClSeI, MeSeI, SIN-2, SIN-3 and SIN-4 ($F_{(6,14)} = 66.55$; $p < 0.0001$; $F_{(6,14)} = 166.5$; $p < 0.0001$; $F_{(6,14)} = 47.73$; $p < 0.0001$;

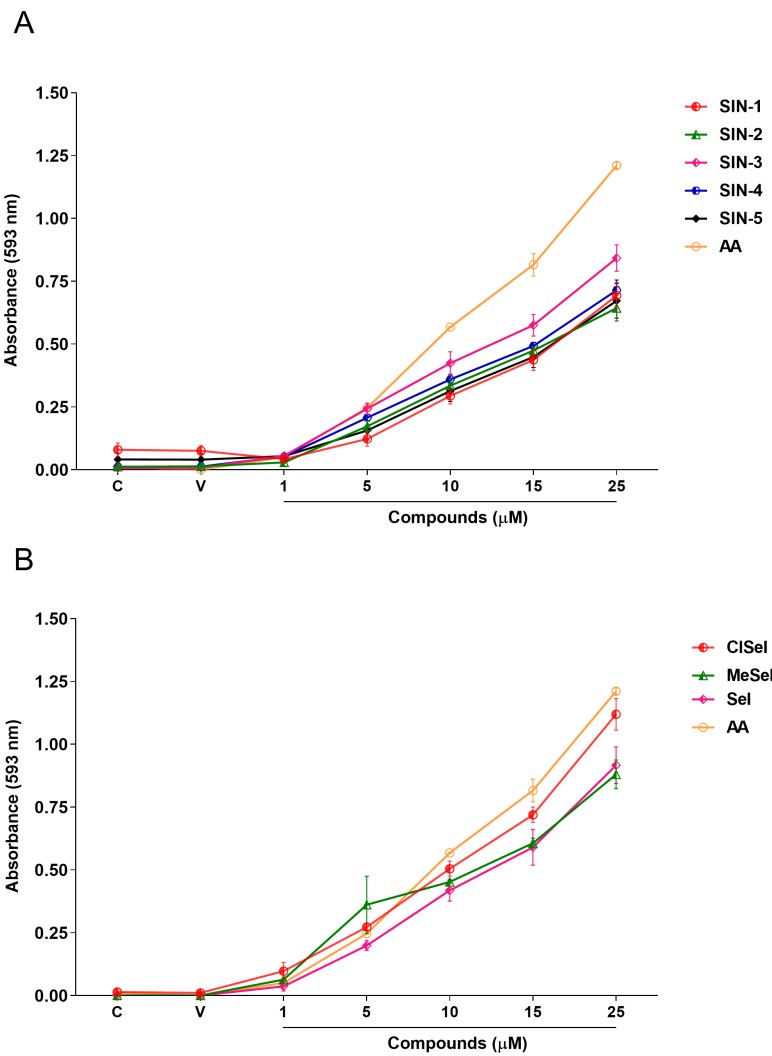

**Figure 3** Effects of (A) thioindolizines and (B) selenoindolizines on the FRAP assay. Ascorbic acid (AA) was employed as a positive control. The data are expressed in mean ± SEM of three distinct experiments conducted in duplicate. The results are presented as a percentage of the vehicle (V) group. FRAP, ferric ion ($Fe^{3+}$) reducing antioxidant power; C, control group.

$F_{(6,14)} = 100.2$; $p < 0.0001$; $F_{(6,14)} = 100.7$; $p < 0.0001$; $F_{(6,14)} = 209.1$; $p < 0.0001$, respectively) compounds presented FRAP activity in concentrations equal to or higher than 5 µM, while the compounds SIN-1 and SIN-5 ($F_{(6,14)} = 52.80$; $p < 0.0001$; $F_{(6,14)} = 37.90$; $p < 0.0001$, respectively) present the FRAP activity in concentrations equal or higher than 10 µM. The positive control AA was effective in ferric-reducing antioxidant power at concentrations equal to or higher than 5 µM when compared to the vehicle group ($F_{(6,14)} = 662.8$; $p < 0.0001$) (Fig. 3).

## Lipid peroxidation induced by SNP

SIN-3 significantly reduced the TBARS levels induced by SNP at concentrations equal or higher to 2 µM ($F_{(8,18)} = 140.5$; $p < 0.0001$). SIN-1, SIN-2 and SIN-5 significantly

reduced the TBARS levels at concentrations equal to or higher than 4 µM ($F_{(8,29)} = 11.07$; $p < 0.0001$; $F_{(8,19)} = 26.27$; $p < 0.0001$; $F_{(8,28)} = 8.21$; $p < 0.0001$, respectively). SIN-4 demonstrated to be effective in reducing TBARS levels at concentrations equal to or higher than 5 µM ($F_{(8,26)} = 10.91$; $p < 0.0001$). All selenoindolizines reduced lipid peroxidation in the homogenate of the mouse brain only at the concentration of 10 µM ($F_{(8,18)} = 35.90$; $p < 0.0001$; $F_{(8,18)} = 5.21$; $p = 0.0018$; $F_{(8,18)} = 36.60$; $p < 0.0001$, SeI, ClSeI and MeSeI, respectively), when compared to the vehicle group. Furthermore, the thioindolizines SIN2, SIN-3 and SIN-4 demonstrated to reduce the TBARS to levels lower than the control group. The same result was observed in the concentration of 5 µM of SIN-2. In the same context, the selenoindolizines SeI and MeSeI also demonstrated this effect at the concentration of 10 µM (Fig. 4). The positive control Trlx slightly reduced the TBARS levels induced by SNP at concentration of 10 µM and reduced the TBARS levels to the control levels only at concentration of 100 µM (Fig. 4).

**Protein carbonylation assay**

Regarding the carbonyl content, one-way ANOVA revealed that SIN-1, SIN-2, SIN-3, SIN-5, SeI, ClSeI and MeSeI ($F_{(6,14)} = 27,85$; $p < 0.0001$; $F_{(6,14)} = 28.64$; $p < 0.0001$; $F_{(6,14)} = 49.14$; $p < 0.0001$; $F_{(6,14)} = 31.18$; $p < 0.0001$; $F_{(6,14)} = 13.43$; $p < 0.0001$, $F_{(6,14)} = 77.30$; $p < 0.0001$; $F_{(6,14)} = 130.5$; $p < 0.0001$, respectively), present significant effects, decreasing the levels of protein carbonylation when compared with the induced group at concentrations equal to or higher than 10 µM. Further, SIN-4 also demonstrated to be effective in reducing the levels of protein carbonylation when compared with the induced group; however, only at concentrations equal to or higher than 25 µM ($F_{(6,14)} = 50.88$; $p < 0.0001$). The thioindolizine SIN-3 and the selenoindolizine MeSeI demonstrated to significantly reduce the protein carbonylation to levels lower than the control group at concentrations equal to or higher than 25 µM. This effect was also observed for the thioindolizines SIN-4 at concentrations equal to or higher than 50 µM and SIN 5, at 100 µM (Fig. 5). The positive control Trlx reduced protein carbonylation induced by SNP in concentrations equal to or higher than 25 µM ($F_{(6,14)} = 4.191$; $p = 0.0128$) (Fig. 5).

## DISCUSSION

It is known that several diseases, such as cardiovascular diseases, mental disorders, and neurodegenerative diseases, are linked to the production of reactive oxygen species (ROS), oxidative stress (Li et al., 2013), excitotoxic events, and membrane permeability changes (Salim, 2017). In this way, multitarget neuroprotective drugs with antioxidant activity have been explored (Gülcan & Orhan, 2020). In the current study, we used the DPPH, FRAP, TBARS, and protein carbonylation assays to demonstrate the antioxidant activity of SIN-1, SIN-2, SIN-3, SIN-4, SIN-5, SeI, ClSeI, and MeSeI.

Antioxidants are agents that can avoid or decrease the free radical damage to cells. Generally, non-enzymatic antioxidants are recognized agents due to their ability to eliminate oxidants through redox reactions, whereby the reactive species are concurrently reduced as another molecule undergoes oxidation (Benzie & Strain, 1996). In this study, a battery of *in vitro* experiments was employed to assess the compounds for potential

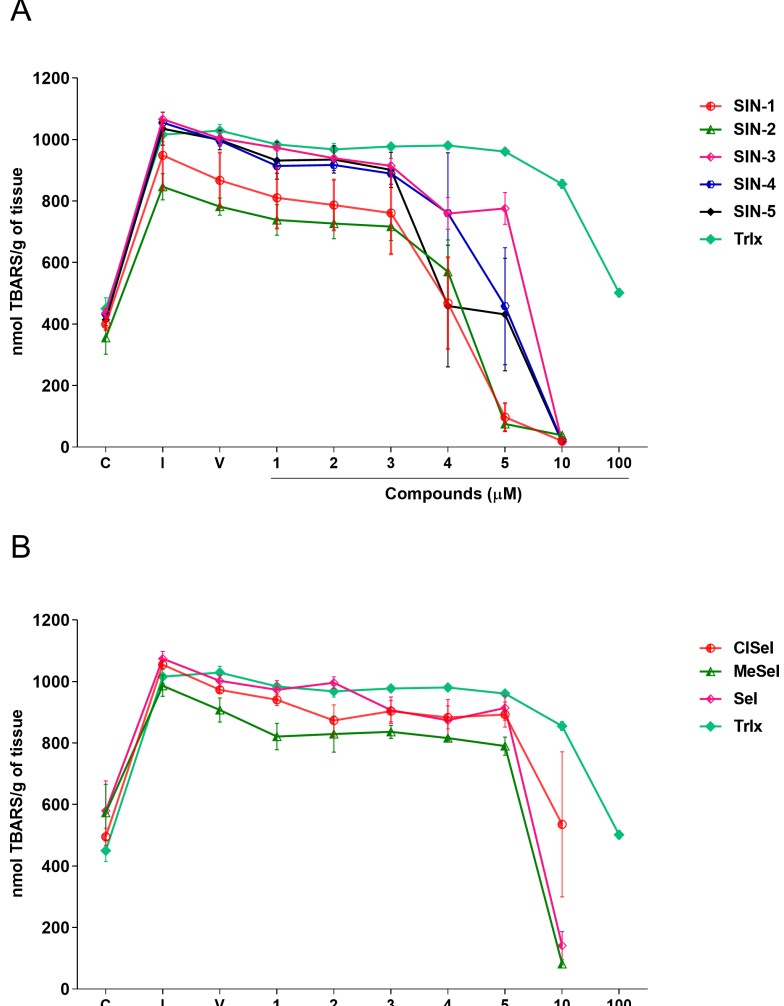

**Figure 4  Effects of (A) thioindolizines and (B) selenoindolizines on the lipid peroxidation assay.**
Trolox (Trlx) was employed as a positive control. The data are expressed in mean ± SEM of three distinct experiments conducted in duplicate. TBARS, thiobarbituric acid reactive species; C, control group; I, induced group; V, vehicle group.

antioxidant properties. Currently, antioxidant evaluation methods are mostly focused on spectrophotometric analyses using hydrogen atom transfer (HAT) and single electron transfer (SET) processes, such as DPPH and FRAP assays, respectively (*Sirivibulkovit, Nouanthavong & Sameenoi, 2018*), and several colorimetric methods have been utilized to assess the reducing potential of both natural and synthetic compounds.

Among them, DPPH is a simple antioxidant screening test that does not require the use of biological tissue. In comparison to natural radicals, DPPH is highly stable and cheap, which makes it ideal for practical and analytical applications (*Yeo & Shahidi, 2019*). Furthermore, in the presence of a molecule that can donate a hydrogen atom, the formation of reduced DPPH is accompanied by the loss of a violet color. Our findings showed that

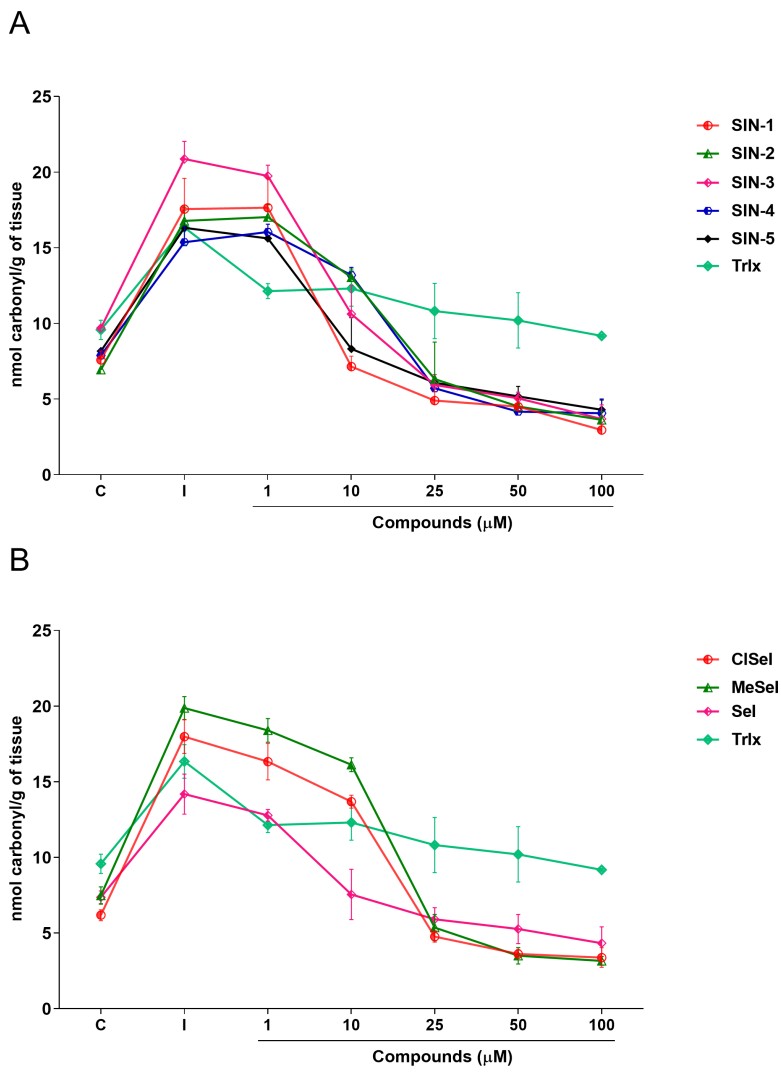

**Figure 5** Effects of (A) thioindolizines and (B) selenoindolizines on the protein carbonylation assay. Trolox (Trlx) was employed as a positive control. The data are expressed in mean ± SEM of three distinct experiments conducted in duplicate. C, control group; I, induced group.

SeI, and SIN-1 had DPPH radical scavenger activity at concentrations of 100 μM or higher, while ClSeI, MeSeI, SIN-3 and SIN-4 revealed this effect only at concentrations equal to or higher than 300 μM, demonstrating that these compounds can scavenge the unpaired electron by a hydrogen-transfer process. Even though statistical significance was achieved for these compounds, compared with the positive control AA, that showed DPPH scavenger activity from the concentration of 10 μM, these compounds demonstrated lower DPPH radical scavenging activity. AA is a well-known antioxidant molecule and the most efficient aqueous-phase antioxidant in human blood plasma, with substantial implications for illness and degenerative processes produced by oxidative stress. Furthermore, SIN-2 and SIN-5 did not show any effect on the radical scavenging activity.

The differences in the antioxidant effect against the synthetic radical DPPH observed for the chalcogen indolizines tested may be related to the structure–activity relationship (SAR) and the subsequent binding of substituents to the phenolic rings, since all compounds containing selenium showed antioxidant activity in the DPPH test, suggesting that the addition of selenium may be responsible for the ability to neutralize the DPPH radical. Moreover, compounds containing the methyl and fluorine groups linked to the phenolic rings have also demonstrated antioxidant activity against the DPPH radical (*Buravlev, Shevchenko & Kutchin, 2021*; *Dinesha et al., 2015*), which may explain the antioxidant action of compounds SIN-3 and SIN-4.

Previously, our research group demonstrated that thioindolizines and ClSeI presented a significant gradual scavenging activity of ABTS from the concentration of 5 µM and the scavenging activity of the compounds was similar to that of the positive control AA (*Penteado et al., 2019*). The differences in the effects of chalcogen-indolizines in the DPPH and ABTS assays can be attributed to the chemical compositions of these two synthetic radicals. Further, the activity can vary depending on the structure of the indolizine compound, such as in SIN-2 and SIN-5 that does not demonstrate effect in DDPH assay, but in the ABTS test the antioxidant activity was revealed (*Penteado et al., 2019*), thus, suggesting that these compounds act *via* electron transfer.

Previous studies also evidenced a lower inhibition of some indolizine derivatives against DPPH radicals (*Narajji, Karvekar & Das, 2008*; *Uppar et al., 2020*). These findings suggest that the antioxidant activity of indolizine compounds may vary depending on the free radical used in the assay, with higher antioxidant activity being observed towards the $ABTS^{\bullet+}$ radical. Furthermore, the DPPH assay is based on an electron transfer reaction and hydrogen-atom abstraction (*Soong & Barlow, 2004*), while the ABTS assay is principally based on a single electron transfer, and the reduction of $ABTS^{\bullet+}$ radical cations can be more efficient than that of DPPH, mainly if a labile hydrogen atom is not present in the compound to be tested (*Prior, Wu & Schaich, 2005*). Once our data revealed that the compounds presented higher radical scavenging efficiency in the ABTS assay when compared to the DPPH one, we suggest that the mechanism of antioxidant activity is principally based on single electron transfer.

In this context, the reducing capacity of the compound may be substantially associated with its antioxidant activity (*Sultana, Anwar & Przybylski, 2007*). In this way, the FRAP assay was utilized to measure the reducing power of the synthesized compounds to better understand the relationship between the antioxidant effect and the reducing power. All compounds revealed a concentration-dependent ferric-reducing ability similar to the positive control AA. Furthermore, the ferric reduction power was observed at lower concentrations than those observed in the antioxidant DPPH experiment. This finding reinforces the hypothesis that the antioxidant activity of these compounds could be mainly due to electron transfer, especially compounds SIN-2 and SIN-5, which showed effects in both the ABTS and FRAP tests (ET-based) but not in the DPPH assay (HAT-based). Once $Fe^{3+}$ can react with peroxides to form radicals and several areas of the brain have high levels of iron (*Victoria et al., 2014*; *Wang & Wang, 2017*), the ferric-reducing ability of the compounds could possibly represent a neuroprotective strategy against some illnesses

caused by oxidative stress. Protein carbonylation is considered a well-established marker of global protein oxidative damage in organisms. Protein oxidation in tissues plays a key role in the aging process and in the pathophysiology of diseases. Oxidative damage in proteins results from various oxidative reactions, causing damage to sulfur-containing, aliphatic and aromatic amino acids (*Georgiou et al., 2018*; *Weber, Davies & Grune, 2015*). In this study, SNP was used as an inductor of protein carbonylation, and all indolizines with selenium or sulfur exhibit antioxidant activity against this damage. Interestingly, SIN-3, SIN-4, SIN-5 and MeSeI were effective in reducing the protein carbonylation to levels lower than the control group, suggesting a high antioxidant activity of these compounds, even better than the positive control Trlx, that reduced the protein carbonylation from the concentration of 25 µM but only to the control levels. Sulfur and selenium compounds have been studied for their antioxidant properties, thus, benefiting the prevention and treatment of chronic diseases (*Kieliszek & Blazejak, 2013*; *Mistry & Williams, 2011*). Studies *in vitro* demonstrated that the organochalcogen compounds significantly inhibited protein carbonylation levels (*Acker et al., 2009*; *Bortolatto et al., 2013*; *Souza et al., 2009*). In relation to the antioxidant activity of the *N*-heterocyclic counterpart, it was demonstrated that 7-chloro-4-(phenylselenyl)quinoline was effective in reducing the protein carbonyl content in mouse brains submitted to the administration of SNP (*Vogt et al., 2018*), corroborating with our findings. Regarding the constituents of the brain, it is well known that lipids constitute approximately 50–60% of its dry weight (*Di Paolo & Kim, 2011*; *Dietschy & Turley, 2001*), and play critical roles in structure and function of this organ. Among the assays that measure the oxidative stress of tissues, TBARS has been widely used in *in vitro* analysis. This test aims to evaluate the reaction of lipid peroxidation products, such as malondialdehyde (MDA), with TBA under acidic and heated conditions. This reaction produces a colored complex that can be quantified spectrophotometrically within a specific wavelength range. Using *in vitro* approaches, our data showed that the compounds evaluated decreased the TBARS levels induced by SNP at low concentrations, compared with the positive control Trlx. Furthermore, compounds SIN-2, SIN-3, SIN-4, SeI and MeSeI reduced TBARS to levels lower than the control group at concentration of 10 µM or 5 µM, in the case of SIN-2. The positive control Trlx slightly decreased TBARS levels at 10 µM and reduced the lipid peroxidation to the control levels at 100 µM. These results reinforce the observed antioxidant potential of these compounds. Our research group recently published data on the antidepressant-like effect of SeI and MeSeI, and their involvement with the serotonergic system (*Garcia et al., 2022*; *da Rocha et al., 2023*). In these studies, it was observed for the first time the antidepressant-like effect of oral acute administration of chalcogen-indolizines, in male Swiss mice. Interestingly, the relation between oxidative stress and depression is well established and some antioxidants have shown antidepressant-like effects in animal models (*Bakunina, Pariante & Zunszain, 2015*; *Bhatt, Nagappa & Patil, 2020*) and, therefore, the antioxidant activity of the chalcogen-indolizines observed in the present study could account for their antidepressant-like effect.

Although there were some small differences already described in the effects of different thio- and selenoindolizines and the respective prototypes without substituents, the

antioxidant effects of the evaluated compounds were quite similar and similar or in some assays even better than the positive controls at same concentrations. These results point these compounds as potential candidates to be tested as therapeutic adjuvants in oxidative stress-related diseases. This study describes the screening protocol for the antioxidant properties of selenoindolizines and thioindolizines using various *in vitro* assays. However, due to the *in vitro* nature of the protocol, there are limitations that prevent a comprehensive understanding of the antioxidant profile of these compounds. Therefore, to better elucidate the mechanism of action of these substances in animals, it is necessary to carry out preclinical induction protocols. Additionally, there is a need to evaluate the toxicity of these compounds, although previous behavioral tests with SeI and MeSeI showed low toxicity in rodents (*Garcia et al., 2022*; *da Rocha et al., 2023*), which provides initial insights into the compounds' safety profile.

## CONCLUSIONS

Our study demonstrated that eight novel chalcogen-indolizines exhibit antioxidant activity. Although the compounds did not show strong activity as DPPH scavengers compared with the positive control AA, they showed FRAP activity similar to AA and reduced protein carbonylation and lipid peroxidation at low concentrations, showing better results than the positive control Trlx. Our results suggest that these compounds could be studied as therapeutic agents for the treatment of brain diseases that involve oxidative stress.

### Funding

This work was supported by the Coordenação de Aperfeiçoamento de Pessoal de Nível Superior–Brasil (CAPES)–Grant Number 001, Fundação de Amparo à Pesquisa do Estado do Rio Grande do Sul (FAPERGS)–Grant numbers 21/2551-0000728-1 and 21/2551-0000614-5, and Conselho Nacional de Desenvolvimento Científico e Tecnológico–Brasil (CNPq). Eder João Lenardão, Cristiani Folharini Bortolatto, and César Augusto Brüning are recipients of the CNPq Fellowship. The funders had no role in study design, data collection and analysis, decision to publish, or preparation of the manuscript.

### Grant Disclosures

The following grant information was disclosed by the authors:
The Coordenação de Aperfeiçoamento de Pessoal de Nível Superior–Brasil (CAPES): 001.
Fundação de Amparo à Pesquisa do Estado do Rio Grande do Sul (FAPERGS): 21/2551-0000728-1, 21/2551-0000614-5.
Conselho Nacional de Desenvolvimento Científico e Tecnológico–Brasil (CNPq).

### Competing Interests

Eder João Lenardão is an Academic Editor for PeerJ.

## Author Contributions

- Cleisson Schossler Garcia conceived and designed the experiments, performed the experiments, analyzed the data, prepared figures and/or tables, authored or reviewed drafts of the article, and approved the final draft.
- Marcia Juciele da Rocha performed the experiments, analyzed the data, prepared figures and/or tables, authored or reviewed drafts of the article, and approved the final draft.
- Marcelo Heinemann Presa performed the experiments, analyzed the data, prepared figures and/or tables, authored or reviewed drafts of the article, and approved the final draft.
- Camila Simões Pires performed the experiments, analyzed the data, prepared figures and/or tables, authored or reviewed drafts of the article, and approved the final draft.
- Evelyn Mianes Besckow performed the experiments, analyzed the data, prepared figures and/or tables, authored or reviewed drafts of the article, and approved the final draft.
- Filipe Penteado performed the experiments, prepared figures and/or tables, authored or reviewed drafts of the article, and approved the final draft.
- Caroline Signorini Gomes performed the experiments, prepared figures and/or tables, authored or reviewed drafts of the article, and approved the final draft.
- Eder João Lenardão conceived and designed the experiments, authored or reviewed drafts of the article, and approved the final draft.
- Cristiani Folharini Bortolatto conceived and designed the experiments, authored or reviewed drafts of the article, and approved the final draft.
- César Augusto Brüning conceived and designed the experiments, analyzed the data, prepared figures and/or tables, authored or reviewed drafts of the article, and approved the final draft.

## Animal Ethics

The following information was supplied relating to ethical approvals (i.e., approving body and any reference numbers):

UFPel Animal Ethics and Welfare Committee (CEEA 12231-2019).

## Data Availability

The raw measurements are available in the Supplementary File.

## Supplemental Information

Supplemental information for this article can be found online at http://dx.doi.org/10.7717/peerj.17074#supplemental-information.

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
