# Peer review of "Exploring the antioxidant potential of chalcogen-indolizines throughout in vitro assays"

_PeerJ, doi:10.7717/peerj.17074_

## Round 0.1 · original submission · Major Revisions

All three reviewers gave suggestions for modification. Please try your best to revise and answer questions from experts.

Reviewer 1 ·

Basic reporting

1. Title. The word “investigations” is imprecise and too general; it includes many actions not considered in this study. I would use “assays”.
- Chalcogen or chalcogeno is the correct term?

2. The original contribution must be clarified since several studies have conducted a series of assays determining the antioxidant potential of chalcogen-indolizines-related compounds in different organs and species (Nobre et al, 2014. DOI: 10.1016/j.bmc.2014.08.018 ;
Al Zahrani et al, 2020. DOI: 10.3390/molecules25194566). Along this manuscript, the chalcogen-indolizines derivatives seem to be the central topic, and their role as antioxidants; however, the author´s description of these compounds, their general procedure of synthesis, and specific structural characteristics are not as specific as expected (please see the above-mentioned studies).

In the introduction section, the authors do not emphasize their contribution to specifically protecting the brain cells, the importance of this specific tissue, and why this needs to be better understood. The contribution of this study is not only in the ability of synthetic compounds as antioxidants but in the possibility of protecting nervous tissue. A clear explanation is needed to support why the authors used brain cells; this should be explained.

Experimental design

3. Methods. Most methods need to be better described.

-The section that includes the Free radical scavenging activity (DPPH) assay section does not describe whether microplates, microtubes, or quartz cells were used for determinations or what kind of spectrophotometer, trade, or model was used. Quantities (mL or microlitres) contained in each reaction? Any negative control?.. please describe.

-Ferric Reducing Antioxidant Power (FRAP) assay
At which wavelength absorbance was determined? Please describe the quantities (mL or microliters) included in reactions?; which was the final volume of each reaction? Please remember that methods should be described enough that any other researcher can reproduce them.

-The molecules/ compounds and some other words refer to the chalcogen-indolizines- derivatives along the manuscript; please always use the same term to avoid confusion. Also, antioxidant potential, power, action, or activity should be analyzed and used correctly.

-Lanes 185-186- 10 mL were added

-Lipid peroxidation assays and protein carbonylation assays also lack information; equipment such as homogenizers, incubators, spectrometers, and centrifuges should be described by trade and models.

-Statistical analysis. Were normal distribution and variances homogeneity determined before ANOVA? ; this should be described.

-Figures 2 /3, 4/5, 6/7, 8/9 should be considered in a single figure with A) and B) sections since the measure the same variable.

-Figure legends should be corrected as they are not as clear as expected. Some abbreviations, such as C and I, are not described and should be included. Signals indicating statistical differences (p < 0.05 and p < 0.001) are confusing, as legends describe.

Validity of the findings

The discussion section needs to be revised since it contains repeated information from the introduction and methods descriptions. The authors should better explain why some compounds could not scavenge radicals from the structural basis.
- Some interpretations about results should be carefully considered since based possibilities that should be better explored in future studies:
“ The ferric-reducing ability of the compounds could possibly represent a neuroprotective strategy against some illnesses caused by oxidative stress”
“The antioxidant activity of the chalcogen-indolizines observed in the present study could account for their antidepressant-like effect”

Additional comments

- The author's future perspectives should propose and support those specific compounds that resulted in better and desirable characteristics to be used in preclinical induction protocols to control oxidative stress damage in the brain.

Reviewer 2 ·

Basic reporting

Garcia et al. conducted a study synthesizing eight S- or Se-substituted indolizines and examining their antioxidative properties in vitro. The topic is intriguing and presents novel aspects. However, the reliance on vehicle controls for statistical comparisons rather than on established positive controls somewhat mutes the demonstration of the antioxidative potential of the compounds. Additionally, a comparative analysis between the S- or Se-substituted indolizines and their original indolizine counterparts is notably absent. The inclusion of in vitro cell-based assays to assess intracellular ROS levels and cytotoxicity of these compounds is also essential. Below are specific areas for improvement.

Experimental design

1. The abstract would benefit from a brief introduction to the indolizine nucleus, explaining why it was chosen as a model molecule for this study.

2. In the introduction, outlining the connection between the diseases of interest and oxidation processes would offer readers a better foundational understanding before delving into the roles of ROS and RNS.

3. To substantiate the purity and structure of the synthesized chemicals, the inclusion of GC-MS and NMR data in the supplementary files is recommended. Furthermore, all the chemical compounds should be clearly abbreviated in each panel of Figure 1 for clarity.

4. Within the Methods section, details regarding the brand and a succinct description of the synthetic radical solution used should be added.

5. The manuscript currently lacks cell-based assays, which are crucial to evaluate the cytotoxicity and intracellular ROS/RNS levels post-treatment with the synthesized compounds.

Validity of the findings

6. For a more comprehensive analysis, statistical comparisons should be made with positive controls (such as AA or Trlx) in addition to the vehicle control group, particularly in the assays conducted throughout the manuscript.

7. In Figure 3, the data points corresponding to AA are missing and should be incorporated.

8. In Figures 4 and 5 (FRAP assay), including the dose-response data of AA is vital for a comparative analysis. Similarly, for Figures 6, 7, 8, and 9, the dose-response data of Trlx should be included for comparison.

9. A crucial aspect of this study is the comparison between the S- and Se-derivatives and their original indolizine counterparts to highlight the benefits of chemical modification. This comparative data is currently missing and should be included to underscore the significance of the study's findings.

Reviewer 3 ·

Basic reporting

Exploring the Antioxidant Potential of Chalcogeno-indolizines throughout in vitro investigations.
This study focuses on investigating the antioxidant properties of eight chemical compounds containing sulfur (S) and/or selenium (Se) with an indolizine nucleus and various substituents. The research aims to explore the potential application of these compounds in treating disorders associated with oxidative stress. Oxidative stress results from the excessive production of reactive oxygen species (ROS) and reactive nitrogen species (RNS), which can lead to cellular and tissue damage, contributing to the aging process and age-related pathologies such as heart diseases and neurodegenerative disorders.

1. BASIC REPORTING
My suggestions/opinions:
Key words: please make Chalcogeno-indolizines nucleus instead indolizine nucleus, I think that in key words should be a material – form example – mice brain
Line 72: which disfunction?
Introdution is well written. This section contains important informations about chalcogeno-indolizines, their derivatives, and its role.
Material & methods – Animal, Chemicals, Free radical scavenging activity (DPPH) assay, Ferric Reducing Antioxidant Power (FRAP) assay, Lipid peroxidation assay, and Protein carbonylation assay is well written. I haven’t got suggestion.
Results –
Line 351 - why all compounds not only reduced the TBARS levels but did so at low concentrations?
Line 366 - the authors stated that due to the in vitro nature protocol, there are limitations that prevent a full understanding of the antioxidant profile of these compounds. Therefore, it is worth learning better about their mechanism of action substances in animals, it is necessary to conduct preclinical induction protocols. Will the authors continue such research?
Conclusion is not enough.
Figures are made in correct way.

Experimental design

2. EXPERIMENTAL DESIGN
The publication is concise and short, but provides necessary information from the research conducted. In my opinion, the experiment was planned correctly. The study uses simple one-way anova statistics. The presentation of the results is also correct.

Validity of the findings

3. VALIDITY OF THE FINDINGS
The research design plays a pivotal role in ensuring internal validity, encompassing the careful selection of compounds, well-defined experimental procedures, and effective control measures to minimize potential confounding variables. The use of established laboratory tests, such as DPPH scavenger activity, FRAP, TBARS, and PC, adds robustness to the measurements, contributing to the accuracy of the findings.

Additional comments

4. General comments
Above.


Good luck!

Annotated reviews are not available for download in order to protect the identity of reviewers who chose to remain anonymous.

---

## Round 0.2 · Minor Revisions

Our Section Editor commented:

(1) I suggest avoiding general "big claims' such as "these compounds have the potential to be used as therapeutic agents for the treatment of brain diseases that involve disturbances in the redox status.". What is the morphological impact of these data?

(2) I did not identify obvious problems in terms of experimental design. However, I do think that data presentation can be improved. Also, consistency in referencing.

(3) Also, the authors cannot make redox claims when they did not investigate redox enzymes like catalase or Glutathiones.

Please address these items i nyour next revision.

Reviewer 1 ·

Basic reporting

Corrected

Experimental design

Corrected

Validity of the findings

Valuable

Additional comments

The manuscript was adequately corrected and has the appropriate publication level.
Some minor errors, such as blank spaces and misspelled words, should be corrected, especially in the discussion section.

Reviewer 2 ·

Basic reporting

The authors have carefully addressed all my comments.

Experimental design

The manuscript is ready for publication.

Validity of the findings

See above,

---

## Round 0.3 · accepted · Accept

The authors have revised the manuscript, and I am not aware of any additional problems.